# Efficient Top-K Identical Frequent Itemsets Mining without Support Threshold Parameter from Transactional Datasets Produced by IoT-Based Smart Shopping Carts

**DOI:** 10.3390/s22208063

**Published:** 2022-10-21

**Authors:** Saif Ur Rehman, Noha Alnazzawi, Jawad Ashraf, Javed Iqbal, Shafiullah Khan

**Affiliations:** 1Department of Computer Science, University of Peshawar, Peshawar 25120, Pakistan; 2Department of Computer Science and Engineering, Yanbu Industrial College, Royal Commission for Jubail and Yanbu, Yanbu Industrial City 41912, Saudi Arabia; 3Institute of Computing, Kohat University of Science and Technology, Kohat 26000, Pakistan; 4Faculty of Computer and Software Engineering, Huaiyin Institute of Technology, Huai’an 233003, China

**Keywords:** IoT, sensors, frequent itemsets mining, artificial intelligence, data mining

## Abstract

Internet of Things (IoT)-backed smart shopping carts are generating an extensive amount of data in shopping markets around the world. This data can be cleaned and utilized for setting business goals and strategies. Artificial intelligence (AI) methods are used to efficiently extract meaningful patterns or insights from such huge amounts of data or big data. One such technique is Association Rule Mining (ARM) which is used to extract strategic information from the data. The crucial step in ARM is Frequent Itemsets Mining (FIM) followed by association rule generation. The FIM process starts by tuning the support threshold parameter from the user to produce the number of required frequent patterns. To perform the FIM process, the user applies hit and trial methods to rerun the aforesaid routine in order to receive the required number of patterns. The research community has shifted its focus towards the development of top-K most frequent patterns not using the support threshold parameter tuned by the user. Top-K most frequent patterns mining is considered a harder task than user-tuned support-threshold-based FIM. One of the reasons why top-K most frequent patterns mining techniques are computationally intensive is the fact that they produce a large number of candidate itemsets. These methods also do not use any explicit pruning mechanism apart from the internally auto-maintained support threshold parameter. Therefore, we propose an efficient TKIFIs Miner algorithm that uses depth-first search strategy for top-K identical frequent patterns mining. The TKIFIs Miner uses specialized one- and two-itemsets-based pruning techniques for topmost patterns mining. Comparative analysis is performed on special benchmark datasets, for example, Retail with 16,469 items, T40I10D100K and T10I4D100K with 1000 items each, etc. The evaluation results have proven that the TKIFIs Miner is at the top of the line, compared to recently available topmost patterns mining methods not using the support threshold parameter.

## 1. Introduction

The IoT is a comprehensive and much researched direction nowadays, and it can be categorized into two main components: embedded systems using sensors and processing technologies on Wi-Fi sensor networks. IoT is evolving day by day, yet it suffers from three main challenges: architecture [1], interoperability [2], and security [3,4,5]. Various IoT-enabled heterogeneous devices around us communicate through sensors via the internet to provide smart services. For example, smart health monitoring systems [6], smart parking systems [7], and smart shopping carts [8]. In this paper, we address one of the pertinent issues with smart shopping carts’ transactional data collected on webservers. After preprocessing this coarse-grained data, some artificial intelligence method can be utilized for effective analysis, such as to study customer buying patterns, effective catalog deign, efficient stock management, etc. [9,10,11]. Frequent itemsets mining is one such method in association rule mining that can serve the aforementioned purpose perfectly. Frequent Itemsets Mining (FIM) is the discovery of those itemsets which coexist in the given dataset more than the user specified limit [12]. The importance of the FIM process can be seen from the fact that it is used for finding customer purchase trends [13], fraud detection in the banking sector [14], most frequent web pages traversal findings [15], health monitoring [16], data profiling systems [17,18,19], and document clustering [20].

Rakesh Agarwal first highlighted FIM in his seminal work as the “Apriori algorithm”. Subsequently, the research community has floated many proposals for the FIM problem. The proposed solutions for FIM encountered two main issues before and after the discovery of Frequent Itemsets (FIs) [21]. First, the user’s selection of the ideal Support Threshold Parameter (STP) before starting the FIM process. Second, the exponential number of FIs generated as a result of the FIM process. This issue is addressed well in different compact FIs-mining procedures that have been proposed. Hence, the selection of an optimum STP for the FIM process is a harder task as it depends on the characteristics of the datasets. Also, the frequentness of an itemset depends on the STP value that is provided by the user. This value is used by the FIM process to isolate frequent patterns from infrequent patterns. If the user chooses a high threshold, there will be fewer itemsets generated, and choosing a low threshold will generate more patterns. Therefore, the user may tune the FIM process with different STP values to discover the desired FIs. The miss tuning of STP can in turn result in two main problems for the FIM process, i.e., effects on the size of the result space and the size of the search space [21].

Many algorithms have been proposed in the area of mining the topmost frequent itemsets [22,23,24,25,26,27,28,29,30,31]. These algorithms are commonly named K-topmost or topmost FIM approaches. Initial topmost FIM algorithms are derivative from the classical user’s-STP-based FIM techniques and inherit all of their demerits, i.e., the Apriori or FP-tree algorithm. There are two main challenges faced by the topmost FIM algorithms. First, the Apriori-based topmost FIM techniques proposed in [23,24,31] have multiple scans, excessive candidate generation, they are search space focused, and have large search space. Second, the FP-tree-based topmost FIM techniques proposed in [22,25,27,28,32] perform at least two scans of the database, suffer from the large search space problem in the case of dense datasets, and have the memory resident construction of FP-tree. In general, the topmost FIs-mining techniques designed based on the aforementioned principles are computationally inefficient [30]. Recent techniques for topmost patterns mining include the mining top-K frequent patterns without support threshold proposed by Salam et al. [30], the top-K identical frequent without support threshold by Rehman et al. [29], and the top-K frequent itemsets mining technique based on equivalence classes by Iqbal et al. [26]. All Apriori-and-FP-growth independent top-K methods mentioned earlier are not viable to perform due to the two main limitations described below.

**a: No top-K specific pruning method:**—All top-K specific methods so far do not use any top-K candidate itemsets pruning technique apart from an automatic support threshold raising strategy. The aforementioned strategy can only determine whether an itemset is frequent or not after computing its support value. On the other hand, we need a pruning mechanism which guarantees that a particular itemset is infrequent without computing its support value.

**b: Computationally intensive methods:**—The Apriori-based top-K methods, the FP-Growth based top-K methods, and the Apriori-and-FP-Growth-free methods are computationally intensive on dense datasets with large value of K. Apriori-and-FP-Growth-free methods produce a large number of candidate itemsets, requiring support computation for every itemset. FP-Growth-based top-K methods are computationally extensive due to the FP-Growth method.

### 1.1. Research Objectives

In this article, we propose a novel topmost frequent itemsets mining algorithm named TKIFIs Miner. The proposed algorithm is not a derivative of the Apriori or FP-Growth techniques, but it uses a recursive best-first-based depth-first traversal technique on search space to discover topmost frequent itemsets. During this traversal, two novel pruning strategies are used to early prune the portion of the search space not required for topmost patterns mining. Subsequently, support of the remaining candidate patterns using an intersection of TIDs approach is calculated. Finally, the support of the pattern is checked against the least support of the patterns available in the top-K list. If the support of the discovered pattern is greater than the least support of the patterns available in the top-K list, it is added to the top-K list; otherwise, the pattern is not added to the top-K list.

### 1.2. Organization of the Paper

The article is organized as follows: Section 2 presents previous work related to this area of research, followed by a discussion of the proposed algorithm and a presentation of a few preliminaries about the topmost frequent itemsets mining problem in Section 3. Additionally, a set of examples are presented to highlight the topmost frequent itemsets mining using k as parameter in place of support threshold. In Section 4, the experimental evaluation of the proposed algorithm and the comparison of the TKIFIs Miner with the current benchmark methods are presented. Finally, Section 5 discusses the achievements of this work and future directions that can be unveiled through topmost frequent patterns mining studies.

## 2. Related Work

The top-K FIM is introduced to avoid user-given threshold parameter tuning. On the other hand, top-K FIM is considered a computationally harder task to perform as compared to support-based FIM. Due to the self-explanatory results, it is applied in different real world application domains such as monitoring users’ activity from their movements data [33], COVID-19 virus strains classification and identification [34], and extraction of metadata from dynamic scenarios [35,36].

Salam et al. for the first time suggested considering natural requirements of STP-free K-most FIs to design a novel topmost-FIM technique that was not derived from the aforementioned classical approaches [30]. This technique mines topmost maximal frequent itemsets and top-K maximal frequent itemsets from the given transactional dataset. According to this approach, a topmost maximal frequent itemset in a dataset is a frequent itemset of a highest support with item length greater or equal to 3. On the other hand, a top-K maximal frequent itemset set is a set of K distinct highest support maximal frequent itemsets. In this approach, first the association ratio of all 2-length itemsets is calculated, followed by the construction of an Association Ratio graph (AR-graph). Subsequently, the AR-graph is applied to construct an all-path-source-destination tree (ASD-tree). Finally, the ASD-tree is used to find the top-K maximal frequent itemset set. Additionally, the Top-K Miner introduced by Rehman et al. [29] does not extend any classic STP-based mining techniques. It performs only one scan of the dataset to construct frequent itemsets of length 2, and during this scan the top-K list is updated with frequent itemsets of lengths 1 and 2. The topmost frequent itemsets of length 2 from the top-K list whose support is greater or equal to the Kth support value in the top-K list are selected for topmost k frequent itemsets mining of lengths greater than or equal to 3. Subsequently, every 2-itemset is iteratively combined with the selected nodes of a Candidate Itemsets Search tree (CIS-tree) to form the itemsets of any length greater than 2. Their support is calculated and adjusted accordingly in the top-K list. The Top-K Miner finds all topmost FIs as per the tuned parameter, but suffers as memory-expensive due to the Candidate Itemsets Search tree (CIS-tree). Moreover, Iqbal et al. presented the TKFIM algorithm which inherits the Apriori mining technique for the discovery of K-topmost FIs [26]. This algorithm performs excessive candidate generation because it uses common itemsets prefixes of the already produced topmost frequent itemsets. The frequencies of the itemsets are generated using the diffset support finding method [37]. Apart from the automatic support threshold raising strategy, the algorithm does not adopt any other pruning strategy to perform search space pruning and to avoid excessive candidate generation. Table 1 presents the discussed topmost FIM techniques and their details.

## 3. Frequent Itemsets Mining

In this section, we will discuss the proposed frequent itemsets mining algorithms, which include three methods: the TK_IFIs_ mining method that mines a user-required number of IFIs from the user-given dataset; the Gen-IFIs method that explores the search space to discover TK-IFIs; and the Candidate-IFIs method that generates the set of candidate itemsets for the newly created IFIs. Additionally, some examples will be presented while explaining the frequent itemsets mining methods to give more clarification to the reader. At the beginning, it is worth starting by presenting some preliminaries and definitions.

### 3.1. Preliminaries and Problem Definitions

The support threshold value provided by the user serves as the border between frequent and infrequent patterns. Our proposed FIM method is not based on a support threshold value supplied by the user. Then, how can our proposed method discriminate between frequent and in-frequent patterns to produce the required number of patterns?

There is an important observation related to the selection of a support threshold as far as our proposed method is concerned. The parameter required by the proposed method is K, which determines the number of topmost frequent patterns that are required to be discovered from search space. The value of the K parameter guides the proposed method for the automatic adjustment of the border between the number of required topmost frequent and infrequent patterns. The support count value of the patterns decides the aforementioned border. This is automatically determined or adjusted by the machine based on K value rather than given by the user. Tuning the value of the K for finding the required number of patterns is simple and unlike a support threshold value which is not dependent on the characteristics of the given dataset. Therefore, based on the above discussion, the following definitions are presented for further groundwork:
Identical Itemsets (IIs): One or more itemsets are called IIs if and only if all of them have the same support count that is $I_{\mathrm{IIs}}=\left\{{\;p}_{i}\;|\;\forall {\;p}_{i}\in X\;⋏\;\mathrm{supp}\left(p_{i}\right)=s\;\right\}$, where s ∈S is the support count of every pattern $p_{i}\;$found in the set of support count S.Top-1 Identical Frequent Itemsets (IFI_1_): One or more itemsets are called IFI_1_ if and only if all of them have same support count and their support is the first highest most support in the set of support count S that is $I_{{\mathrm{IFI}}_{1}}=\left\{{\;p}_{i}\;|\;\forall {\;p}_{i}\in X\;⋏\;\mathrm{supp}\left(p_{i}\right)=s_{1}\;\right\}$, where $s_{1}\in S$ is the first highest support count in the set of support count S.Top-K Identical Frequent Itemsets ($I_{{\mathrm{IFI}}_{K}}$): One or more itemsets are called IFIs if and only if all of them have same support count and their support is the Kth highest most support in the set of support count S that is $I_{{\mathrm{IFI}}_{K}}=\left\{{\;p}_{i}\;|\;\forall {\;p}_{i}\in X\;⋏\;\mathrm{supp}\left(p_{i}\right)=s_{K}\;\right\}$, where $s_{K}\in S$ is the Kth highest support count in the set of support count S.


Suppose I is the set of all items in a database, i.e., $I=\left\{i_{1},i_{2},i_{3},\ldots \;\ldots \;\ldots \;\ldots ,i_{n}\right\}$, where i is an item. The database $D=\left\{t_{1},t_{2},t_{2},\ldots \;\ldots \;\ldots \;\ldots ,t_{m}\right\}\;$ is a set of m transactions where every t is a transaction. An itemset p⊆I containing one or more transactions presents support of p, i.e., $\mathrm{Supp}\left(p\right)$. Let X be a set of all $p_{s}$ found in transactions of database D. That is, X contains all the itemsets that can be generated from D. Let S be the set of support counts of all the itemsets contained by X. The problem of topmost frequent itemsets mining is to find all IFIs of highest support to Kth support where K is a user specified number of topmost IFIs, i.e., ${\mathrm{TK}}_{\mathrm{IFI}}=\left\{{\;\mathrm{IFI}}_{1},{\;\mathrm{IFI}}_{2},{\mathrm{IFI}}_{3}\ldots \;\ldots \;\ldots \;\ldots \;\ldots ,{\mathrm{IFI}}_{K}\right\}$.

### 3.2. TK_IFIs_ Mining Method

In this section we introduce ${\;\mathrm{TK}}_{\mathrm{IFIs}}\mathrm{Miner}$ method that mines the user required number of IFIs from the user-given dataset D basd on one simple parameter K as shown in step 1 in the above Algorithm 1. The database is scanned once to copy the tid in the corresponding position of the 1-itemset indexed with item i. It is done by performing the horizontal scan of the given datasets as shown in step 3 to 5 of the above Algorithm 1. The next step is copying all 1-itemsets of the highest support to the Kth support into set of candidate itemset C, and into the TK_IFIs_ list. In order to perform the aforesaid task, first topmost highest support itemset is discovered and copied into itemset i, followed by copying this itemset in C as shown in step 6a and step 6b of the above Algorithm 1. The topmost selected itemset i in step 6a is copied into the TK_IFIs_ list in step 6c of the above Algorithm 1.This itemset i is removed from the list of 1-itemsets in step 6d, as shown above in Algorithm 1, In step 7 of the above Algorithm 1, we used 1-itemsets of highest frequency to Kth frequency for producing frequencies of 2-itemsets. In step 8, we use the ${\mathrm{Gen}}_{\mathrm{IFIs}}$ algorithm for the top-K IFIs mining from the already trimmed search space. For more explanation, we give Example 1 to illustrate the algorithm by applying it to a transactional dataset that is given in Table 2.


**Algorithm 1.**
TK_IFIs_ Miner (Data Set: D, User Input K).


$Candidate\;Itemset:\;C\;\;\;\;\;\;Set\;of\;1-Itemsets:\;1\_itemsets,\;Set\;of\;2-Itemsets:2\_itemsets\;transaction:tr\;Top-K\;IFIs\;List\;Structure:\;TK_{IFIs}\;\mathrm{Steps}$

(1)input K,D // input the number of TK_IFIs_ to mined from dataset D.(2)

$TK_{IFIs\;}=\emptyset ,\;C=\emptyset ,\;IFI=\emptyset ;$

(3)

foreach transaction tr ∈D

(4)

foreach item i ∈tr

(5)

$1\_itemsets_{i}=1\_itemsets_{i}.\;tid\;\cup tr\left(i\right).tid;$

(6)

$Repeat\;Until\;\left|TK_{IFIs}\right|!=K\;//\;\mathrm{copy}\;\mathrm{top}-K\;1-\mathrm{itemset}\;\mathrm{into}\;{\mathrm{TopK}}_{\mathrm{IFIs}}\;\mathrm{list}\;\mathrm{structure}$

     a.

$i=\max\_SupportItem\left(1\_itemsets\right);$

     b.

$C=C\;\cup \;1\_itemsets_{i};$

     c.

$Append\;\left(TK_{IFIs},\;1\_itemsets_{i}\right);$

     d.

$1\_itemsets=1\_itemsets-1\_itemsets_{i};$


(7)

$foreach\;i:\;1\;to\;\left|C\right|-1\;//\;2\text{-}\mathrm{itemsets}’s\;\mathrm{support}\;\mathrm{used}\;\mathrm{for}\;\mathrm{pruning}\;\mathrm{purpose}\;\mathrm{in}\;{\mathrm{Gen}}_{\mathrm{IFIs}}$

     a.

$sprt=support\left(C_{i},\;C_{i+1}\right);$

     b.

$2\_itemsets\left[C_{i},\;C_{i+1}\right]=sprt;$


(8)

$Gen_{IFIs}\left(IFI=\emptyset ,\;C,\;TK_{IFIs},\;K,\;2\_itemsets\right);$

(9)

$return\;TK_{IFIs};$




**Example 1:** Consider the transactional dataset given in Table 2. The dataset consists of 10 transactions and 6 distinct items. For the given value of K = 4, 5, the dataset is mined for top-4, top-5 IFIs as shown in Table 3.

Table 3 represents the topmost patterns mined for K= 4, and 5, from the transactional dataset given in Table 2. It is clear from Table 3 that the only input required from the user is the K value to produce the desired number of topmost patterns. In the top-K result sets in Table 3, it can be observed that all the subsets of an FI in top-K IFIs are placed at the same, or ranked at higher, positions due to Apriori property [1]. For example, all the subsets of a frequent itemset {FBA = 7} in top-5 IFIs in Table 3 are ranked at top-4 or higher positions.

### 3.3. $TK_{IFIs}$ Production Method

This section presents the ${\mathrm{Gen}}_{\mathrm{IFIs}}$ method which is a search space exploration algorithm that discovers the ${\mathrm{TK}}_{\mathrm{IFIs}}$ based on the number of topmost-K IFIs required by the user. It is a recursive best-first-based depth-first search ${\mathrm{TK}}_{\mathrm{IFIs}}$ discovery approach. Every time ${\mathrm{Gen}}_{\mathrm{IFIs}}$ is called recursively, it discovers one or more IFIs. The generated IFIs are then either accommodated in the currently maintained ${\mathrm{TK}}_{\mathrm{IFIs}}$ list or dropped due to lesser support count. The ${\mathrm{Gen}}_{\mathrm{IFIs}}$ procedure uses four parameters during the ${\mathrm{TK}}_{\mathrm{IFIs}}$ discovery process. The first parameter is current highest support${\;\mathrm{IFI}}_{o}$. At the start, when${\;\mathrm{Gen}}_{\mathrm{IFIs}}\;$is called, it is passed as empty. The second parameter is candidate itemsets list $C_{o}$ which is used for the extension of current${\;\mathrm{IFI}}_{o}$. Before calling ${\mathrm{Gen}}_{\mathrm{IFIs}}$ for the first time, the top-K 1-itemsets are copied in $C_{o}$, and subsequently it is passed to ${\mathrm{Gen}}_{\mathrm{IFIs}}$ as the second parameter. The parameter ${\mathrm{TK}}_{\mathrm{IFIs}}$ maintains the current top-K IFIs produced before calling and within the ${\mathrm{Gen}}_{\mathrm{IFIs}}$ calls. The 2−itemsets is a last parameter used in${\;\mathrm{Gen}}_{\mathrm{IFIs}}$ calling. This parameter is used in pruning the candidate patterns in the Candidate_IFIs procedure in step 5 of Algorithm 2.

The ${\mathrm{Gen}}_{\mathrm{IFIs}}$ procedure extends the current IFI with the item i from the candidate itemset list $C_{o}$ iteratively as shown in step 1 of Algorithm 2. The minimum support threshold is raised automatically with the addition of IFIs in the ${\mathrm{TK}}_{\mathrm{IFIs}}$ list. Therefore, before appending item i from the candidate list to the current IFI, its support count is checked with minimum support of the ${\mathrm{TK}}_{\mathrm{IFIs}}$ list, as shown in step 2 of Algorithm 2. If the support of item i is less than the minimum support of IFIs in the ${\mathrm{TK}}_{\mathrm{IFIs}}$ list, then the current item i is pruned from combining it with the current IFI. In step 3, the selected item i is combined with current ${\;\mathrm{IFI}}_{o\;}$ to form a new IFI,${\;\mathrm{IFI}}_{n}$. In step 4, all the 1-itemsets in $C_{o}$ whose support value is greater than or equal to the current item i are selected/copied in the possible candidate itemset list${\;P}_{c}$.

The Candidate_IFIs procedure is used here to return a new candidate list $C_{n}\;$for the new IFI, ${\mathrm{IFI}}_{n}$, as shown in step 5 of Algorithm 2. It uses the new IFI, ${\mathrm{IFI}}_{n}$, formed in step 3, and possible candidate list $P_{c}$, to generate a new candidate itemsets list, $C_{n}$. In step 6, the ${\mathrm{Gen}}_{\mathrm{IFIs}}$ algorithm is called recursively for further processing. The used search space strategy is explained below. For FIM, the search space is arranged in the form of a Set Enumeration tree (SE-tree), as presented in Figure 1. Since FIM methods discover patterns or itemsets in a search space, the SE-tree is one of the search space presentation methods used here to enumerate all of the possible subsets. The intuitions behind adopting the SE-tree for search space presentation is given below.
The problems where search is a subset of power set. It represents irredundant complete search space.The SE tree is a structure that can be traversed recursively.For the FIM techniques adopting depth-first traversal of the search space, SE-tree can be used to model their search space efficiently.



**Algorithm 2.**

${\mathrm{Gen}}_{\mathrm{IFIs}}({\mathrm{IFI}}_{o},{\;C}_{o},{\;\mathrm{TK}}_{\mathrm{IFIs}},\;2-\mathrm{itemsets})$



$\mathrm{Input}\;IFI_{o}:\;\mathrm{Old}\;\mathrm{IFI}\;(\mathrm{current}\;\mathrm{node}\;\mathrm{head})\;C_{o}:\;\;\;\;\mathrm{Old}\;\mathrm{candidate}\;\mathrm{itemset}\;\mathrm{list}\;(\mathrm{current}\;\mathrm{node}\;\mathrm{tail})\;IFI_{n}:\;\mathrm{New}\;\mathrm{IFI}\;(\mathrm{new}\;\mathrm{node}’s\;\mathrm{head}\;\mathrm{portion})\;C_{n}:\;\;\mathrm{New}\;\mathrm{candidate}\;\mathrm{itemset}\;\mathrm{list}\;(\mathrm{new}\;\mathrm{node}’s\;\mathrm{tail}\;\mathrm{portion})\;TK_{IFIs}:\;\;\;\;\mathrm{Current}\;\mathrm{list}\;\mathrm{of}\;\mathrm{Top}\text{-}K\;\mathrm{IFIs}\;\mathrm{discovered}\;\mathrm{so}\;\mathrm{far}\;2\text{-}\mathrm{itemsets}:\;\mathrm{set}\;\mathrm{of}\;2\text{-}\mathrm{items}\;\mathrm{formed}\;\mathrm{from}\;K\;\mathrm{highest}\;\mathrm{support}\;1\text{-}\mathrm{itemset}\;\mathrm{for}\;\mathrm{pruning}\;\mathrm{purpos}\;P_{c}:\;\mathrm{Possible}\;\mathrm{candidate}\;\mathrm{itemset}\;\mathrm{list}$

(1)

$foreach\;item\;i\;\in C_{o}\;\;//\;select\;item\;i\;with\;highest\;support\;count\;from\;C_{o}$

(2)

$if(sprt\left(i\right)<\min\_sprt\left(TK_{IFIs}\right))$



continue; // prune selected item from the list

(3)

$IFI_{n}=IFI_{o\;}\;U\;\left\{i\;\right\};$

(4)

$P_{c}=\left\{\;\forall \;x:\;x\in C_{o}\;and\;support\left(x\right)\leq support\left(i\right)\;\right\};$

(5)

$C_{n}=Candidate\_IFIs\left(IFI_{n},\;P_{c},\;TK_{IFIs},\;2-itemsets\right);$

(6)

$Gen_{IFIs}\;(IFI_{n},\;C_{n},\;TK_{IFIs},\;2-itemsets);$

(7)

$\mathrm{Return}\;TK_{IFIs};$




We now give Example 2 to illustrate the ${\mathrm{Gen}}_{\mathrm{IFIs}}$ method by applying it to the transactional dataset that was given in Table 2.

**Example 2:** Consider the transactional dataset Table 2. Generate all 1-itemset for K = 5 and corresponding 2-itemsets from it. The given transactions dataset is scanned transaction by transaction using iterative step 3 of the Algorithm 1. Every transaction is scanned from left to right for every item. The transaction ID for every item is recorded. Finally, the K highest support 1-itemsets are selected iteratively as shown in steps 6a and 6c of Algorithm 1. The 2-itemsets are then computed from the K highest support 1-itemstets, as shown in the table below.

### 3.4. Candidate Generation Method

The method is used to generate the set of candidate itemset C for the newly created${\;\mathrm{IFI}}_{n}$. Every item from this set will be used subsequently to extend the current${\;\mathrm{IFI}}_{n}$. For the creation of candidate itemset C, individual items from possible candidate list $P_{c}$ are utilized. An item y is iteratively selected from $P_{c}$ in step 1, as shown above in Algorithm 3, for the purpose of addition into candidate list C. Before appending an item y into candidate list C, two types of pruning strategies are applied on it. These strategies are explained in the following.

**Itemset Based Candidate Pruning (IBCP):** An itemset ${\mathrm{IFI}}_{n}\;\cup y$ is not a candidate for IFI if an item in an itemset has support count less than minimum support of IFIs in${\;\mathrm{TK}}_{\mathrm{IFIs}}$.

**Two-Itemset Based Candidate Pruning (TIBCP):** An itemset ${\mathrm{IFI}}_{n}\;\cup y$ is not a candidate for IFI if any 2-itemset in an itemset has support count less than minimum support of IFIs in${\;\mathrm{TK}}_{\mathrm{IFIs}}$.

Let $X\;\subseteq {\mathrm{IFI}}_{n}\cup y$, where X is a two-itemset and y∈X is a one-itemset to extend${\;\mathrm{IFI}}_{n}$. It is clear that the support count of y or X is less than minimum support of IFIs in${\;\mathrm{TK}}_{\mathrm{IFIs}}$, i.e., $\min\_\mathrm{sprt}\left({\;\mathrm{TK}}_{\mathrm{IFIs}}\right)$. This implies that the $\mathrm{sprt}\left({\mathrm{IFI}}_{n}\cup y\;\right)<\min\_\mathrm{sprt}\left({\;\mathrm{TK}}_{\mathrm{IFIs}}\right)$ according to the Apriori property [12]. Hence, an itemset ${\mathrm{IFI}}_{n}\cup y$ can never be termed as IFI.

The IBCP and TIBCP strategies are applied on line numbers 2 and 4, respectively, in Algorithm 3. If the support of an item y, or two-itemset containing an item y, is less than minimum support of IFIs in ${\;\mathrm{TK}}_{\mathrm{IFIs}}$, the item y is skipped from frequency test.


**Algorithm 3.**

$\mathrm{Candidate}\_\mathrm{IFIs}({\mathrm{IFI}}_{n},{\;P}_{c},{\;\mathrm{TK}}_{\mathrm{IFIs}},\;2-\mathrm{itemsets})$



$\mathrm{Input}\;IFI_{n}\;\;:\;\mathrm{New}\;\mathrm{IFI}\;(\mathrm{new}\;\mathrm{node}\;\mathrm{head}\;\mathrm{portion})\;P_{c}\;\;\;:\;\;\mathrm{Possible}\;\mathrm{candidate}\;TK_{IFIs}\;\;:\;\;\mathrm{Current}\;\mathrm{list}\;\mathrm{of}\;\mathrm{Top}\text{-}K\;\mathrm{IFIs}\;\mathrm{discovered}\;\mathrm{so}\;\mathrm{far}\;2\text{-}\mathrm{itemsets}:\;\mathrm{set}\;\mathrm{of}\;2\text{-}\mathrm{items}\;\mathrm{formed}\;\mathrm{from}\;K\;\mathrm{highest}\;\mathrm{support}\;1\text{-}\mathrm{items}\;\mathrm{for}\;\mathrm{pruning}\;\mathrm{purpose}\;\mathrm{Steps}$

(1)

$foreach\;item\;y\;\in P_{c}$

(2)

$if(sprt\left(y\right)<\min\_sprt\left(TK_{IFIs}\right))$

(3)    

continue; // prune selected item from the list

(4)

$if(2-itemset[\;IFI_{n}\left[\left|IFI_{n}\right|\right],y\;]<\min\_sprt\left(TK_{IFIs}\right)\;\vee \;//\;\mathrm{last}\;\mathrm{item}\;\mathrm{of}\;\mathrm{IFI}\;\mathrm{with}\;y$

(5)    

$\;2-itemset\left[\;IFI_{n}\left[\left|IFI_{n}\right|-1\right],y\right]<\min\_sprt\left(TK_{IFIs}\right)\;\vee$

(6)    

 …………………………

(7)    

$\;2-itemset\left[\;IFI_{n}\left[2\right],y\right]<\min\_sprt\left(TK_{IFIs}\right)\;\vee$

(8)    

$\;2-itemset\left[\;IFI_{n}\left[1\right],y\right]<\min\_sprt\left(TK_{IFIs}\right))$

(9)    

 continue; // prune selected item from the list

(10)    

$\;if\left(sprt\left(IFI_{n}\;\cup y\right)\geq \;\min\_sprt\left(TK_{IFIs}\right)\right)$

(11)    

$Append\;\left(TK_{IFIs},IFI_{n}\;U\;y,\;sprt\left(IFI_{n}\;U\;y\right)\right);$

(12)    

C=C∪y;

(13)

Return C;




The support count of the ${\mathrm{IFI}}_{n}\cup y\;$is computed in step 10 of Algorithm 3 as shown above. If the support count of the ${\mathrm{IFI}}_{n}\cup y$ is found equal or greater than the minimum support of any IFI in a set of IFIs, i.e., ${\mathrm{TK}}_{\mathrm{IFIs}}$, the resultant IFI is added to ${\mathrm{TK}}_{\mathrm{IFIs}}$ in step 11 of Algorithm 3 above. The currently selected item y is added to the set of candidates C in step 12 of Algorithm 3 above for further depth-first order processing of itemsets. We now give Example 3 to illustrate the Candidate_IFIs method by applying it to the transactional dataset that was given in Table 2.

**Example 3:** For the given transactional dataset in Table 2, assuming descending support order < on top-5 IFIs as shown in Table 4, the complete set enumeration tree is given above in Figure 1.

If we have n items in a set of items $I=\left\{i_{1},i_{2},i_{3},\ldots \;\ldots \;\ldots \;\ldots ,i_{n}\right\}$ in a database D, the search space for the FIM problem consists of all possible subsets of I, that is $2^{n}$ itemsets. Assuming specific order on the items of set I, the set enumeration tree of I will present all $2^{n}$ itemsets. All nodes of the tree except the root node shown in the above Figure 1 are made according to the following two observations.Let $f_{p}$ be a node label in the SE-tree. Let C⊆I be a candidate set of items which is used to extend any node in the tree apart from root node, that is$\;C=\{z\;|\;z\in I\;\wedge \;\mathrm{supp}\left(z\right)\leq \mathrm{supp}\left(x\right),\;\forall x\;\in f_{p}\;\}$.A node label $f_{c}$ is an extension of a node label $f_{p}$ if $f_{c}=f_{p}\cup X$ where X⊆C, and X is any possible subset of C such that X≠∅. An itemset $F=f\;\cup \left\{z\right\}$ is a single item extension or child of f if z∈C.


## 4. Comparative Evaluation

In this section, we will start by presenting Example 4 and considering the transactional dataset given in Table 2.

**Example 4:** All methods that are included in the frequent itemsets mining algorithm and given in Algorithms 2 and 3 will be applied, in addition to finding all the top-5 *IFIs* from the given dataset, as illustrated below in Table 5, Table 6, Table 7, Table 8 and Table 9. Moreover, the SE-tree presented in Figure 2 is used to enumerate all of the possible subsets.

### Performance Trends

To analyze the performance trends of the${\;\mathrm{TK}}_{\mathrm{IFIs}}\mathrm{Miner}$, it was compared against two recent top-most IFIs mining techniques, top-K Miner [29] and TKFIM [26]. The top-K Miner uses depth-first traversal and is not an Apriori-inspired method, while the TKFIM method uses breadth-first traversal strategy, which is considered an Apriori-inspired approach. All these techniques need one parameter, i.e., K, to find top-most IFIs from the given dataset. For the comparative evaluation, we used six benchmark datasets, as shown in Table 10 below.

The first two datasets shown in Table 10 above are freely downloadable from the UCI Machine Learning Repository [https://archive.ics.uci.edu/ml/index.php (accessed on 15 September 2022)] [38]. The third, fourth, and fifth datasets are freely downloadable from a frequent itemsets mining datasets repository [http://fimi.uantwerpen.be/data/ (accessed on 22 June 2021)] [39]. The T40I10D100K and T10I4D100K datasets are synthetic in nature, generated on IBM Synthetic Data Generator by IBM Almaden Quest research group. The Chess and Connect datasets are dense in nature. Hence, these datasets will generate result patterns of long and short length if K value is large. The Retail, T40I10D100K, and T10I4D100K datasets are sparse in nature and the average presence of items in every transaction is low. These datasets will result in patterns of short length even if a large value of the parameter K is supplied.

For experimental evaluation with TKIFIs Miner, we selected three frequent itemsets mining methods: FP-Growth [40], Top-K Miner [29], and TKFIM [26]. The FP-Growth method is used as a benchmark method for experimental evaluation of the many mining methods. Apart from the FP-Growth method, all the other methods apply K as parameter to find topmost IFIs of highest support to the Kth distinct support. The FP-Growth method uses support threshold value for frequent patterns mining. Additionally, FP-Growth, top-K Miner, and TKIFIs Miner use tree data structure and depth-first strategy for mining patterns, whereas the TKFIM approach is an Apriori-inspired algorithm and uses breadth-first strategy to explore the search space. In the experimental work, the support of Kth IFI will be applied to find patterns with FP-Growth. The mapping of K and support threshold parameter that is used to find the same patterns is already established in the work of the Top-K Miner algorithm [29].

On dense datasets such as Chess, Connect, and T40I10D100K, the performance trends of all the methods are almost similar for small values of K from 1 to 20. The top-K Miner, TKFIM, and TKIFIs Miner arrange the search space in descending support order. Therefore, for small values of K these methods compute IFIs of K-highest support in less amount of time. On the other hand, a high support threshold value enables the FP-Growth method to prune the search space and find the same result efficiently. For large values of K, the performance of the TKIFIs Miner is much better due to IBCP and TIBCP pruning strategies. These pruning strategies play a pivotal role enabling the TKIFIs Miner to prune the entire subtree before even finding the support of the candidate patterns. TKFIM and top-K Miner do not apply pruning strategies, hence they perform excessive candidate generation and support computation of patterns which are not even frequent. Therefore, these methods need more time on dense datasets with large values of K than does the TK**_IFIs_** Miner. For large values of K, the FP-Growth method uses low support threshold to create lots of conditional pattern trees on dense datasets and for a large number of long and short patterns. Hence, it consumes more time than all the top-K methods. Figure 3, Figure 4 and Figure 5 present the comparative results from applying the four approaches, FP-Growth, Top-K Miner, TKFIM, and TK**_IFIs_** Miner, to the Chess, Connect, and T40I10D100K datasets, respectively.

On sparse datasets like Retail and T10I4D100K, the performance of TKIFIs Miner is almost equal for small values of K, as shown in Figure 6 and Figure 7, whereas on large values of K the performance of TKIFIs Miner is better than TKFIM, top-K Miner, and FP-Growth. As already mentioned, the high value of K increases candidate itemsets generation, and the pruning methods start pruning the entire subtrees in TKIFIs Miner, reducing its execution time in comparison to counterparts.

## 5. Conclusions and Future Work

In this article, we presented an efficient TK_IFIs_ Miner algorithm for mining top-K IFIs without using the support threshold parameter from transactional data collected through smart shopping carts. Hence, adjusting this parameter to mine the required number of FIs is a harder choice for users. Therefore, the TK_IFIs_ Miner algorithm allows users to control the production of the number of IFIs using a parameter K. In TK_IFIs_ Miner, as part of an automatic adjustment of the support threshold, we used new IBCP and TIBCP pruning techniques which were not used earlier in any topmost frequent patterns mining problem. These pruning methods have proven the effectiveness of pruning the entire itemsets’ sub-trees from the search space, and hence enabling the depth-first search based on the ${\mathrm{Gen}}_{\mathrm{IFIs}}$ method to quickly find those itemsets having a high support value.

TK_IFIs_ Miner outperformed all the topmost patterns mining approaches and the FP-Growth technique in the experimental evaluation. TK_IFIs_ Miner’s computational time is equal to its counterparts on dense and sparse datasets with small values of K. On dense datasets, TK_IFIs_ Miner excels by a bigger margin in terms of computational time for high values of K. As future work to show the created result-set in a more compact manner, we will integrate maximal or closed patterns mining with the TK_IFIs_ Miner approach.

## Figures and Tables

**Figure 1 sensors-22-08063-f001:**
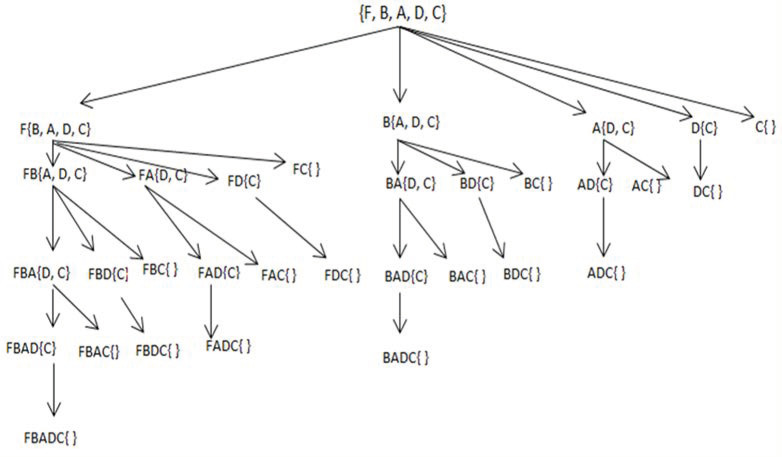
Set enumeration tree for 5 items.

**Figure 2 sensors-22-08063-f002:**
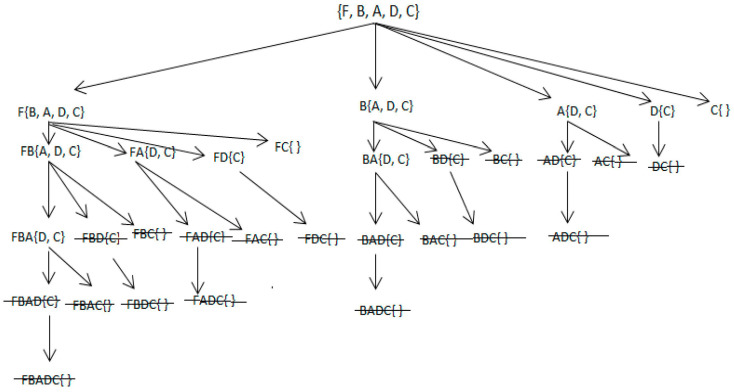
Depth-first-based search space traversal for top-5 IFIs.

**Figure 3 sensors-22-08063-f003:**
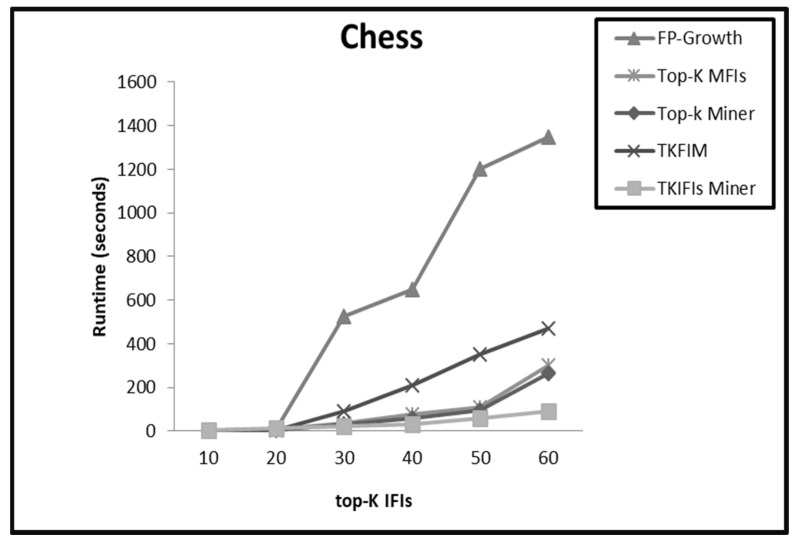
The comparative results from applying the four approaches FP-Growth, Top-K Miner, TKFIM, and TK**_IFIs_** Miner to Chess dataset.

**Figure 4 sensors-22-08063-f004:**
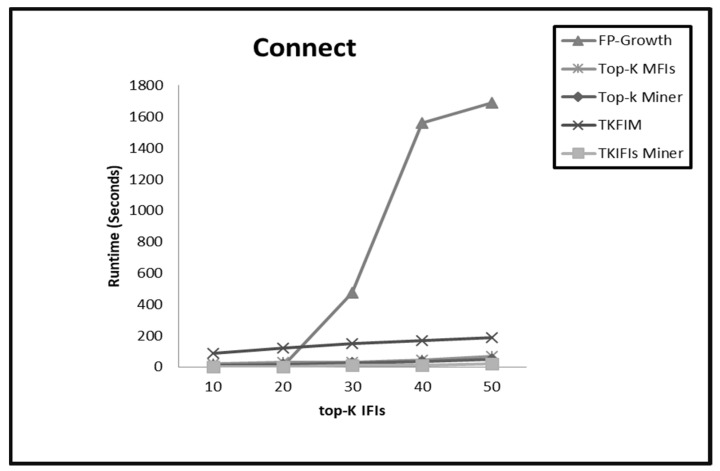
The comparative results from applying the four approaches FP-Growth, Top-K Miner, TKFIM, and TK_IFIs_ Miner to Connect dataset.

**Figure 5 sensors-22-08063-f005:**
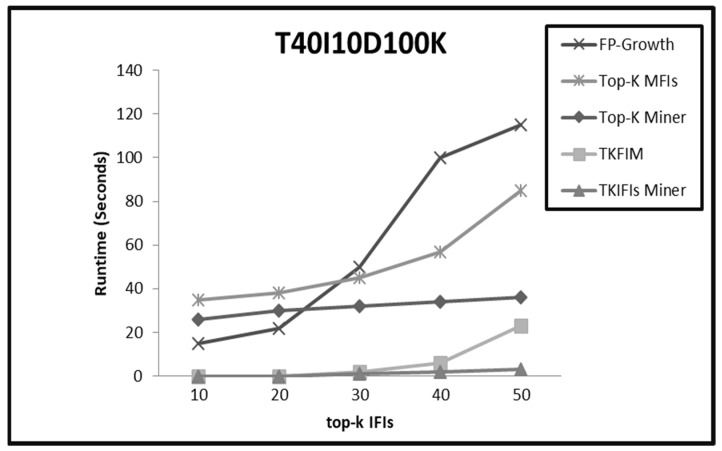
The comparative results from applying the four approaches FP-Growth, Top-K Miner, TKFIM, and TK_IFIs_ Miner to T40I10D100K dataset.

**Figure 6 sensors-22-08063-f006:**
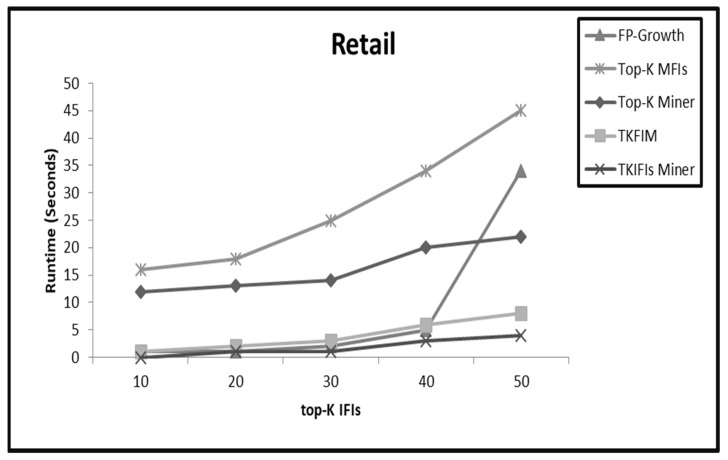
The comparative results from applying the four approaches FP-Growth, Top-K Miner, TKFIM, and TK_IFIs_ Miner to Retail dataset.

**Figure 7 sensors-22-08063-f007:**
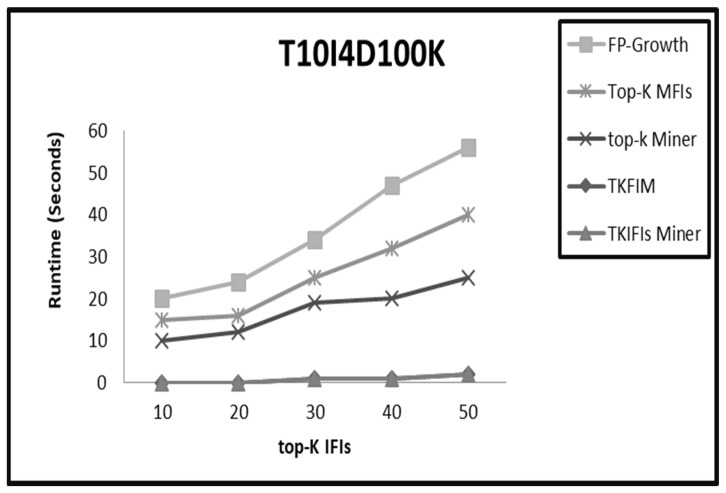
The comparative results from applying the four approaches FP-Growth, Top-K Miner, TK_FIM_, and TK_IFIs_ Miner to T10I4D100K dataset.

**Table 1 sensors-22-08063-t001:** Topmost FIM algorithms and their details.

No	Algorithm Name	Pruning Strategy	Candidate Generation	Technique’s Type	All Patterns
1	Itemset-Loop [24] MTK [23] Top-N [31]	Auto STP raising	Yes	Apriori	All patterns
2	BOMO [32] COFI [27] CRMN [22] ExMiner [28] TFP [25]	Auto STP raising	No	FP-Growth	All patterns
3	Slam [30]	No STP	No	Graph-based approach	Few patterns
4	Top-K Miner [29]	Auto STP raising	Yes	No	All patterns
5	TKFIM [26]	Auto STP raising	Yes	Apriori	All patterns

**Table 2 sensors-22-08063-t002:** Transactional dataset.

TID	Transactions
1	A B D F
2	A B F
3	A D F
4	B C D E F
5	B C D E F
6	A B F
7	A B D F
8	A B F
9	A B C D F
10	A B C E F

**Table 3 sensors-22-08063-t003:** Top-4 IFIs, Top-5 IFIs.

Top-4 IFIs	Top-5 IFIs
(1) F = 10	(1) F = 10
(2) B = 9, BF = 9	(2) B = 9, BF = 9
(3) A = 8, AF = 8	(3) A = 8, AF = 8
(4) AB = 7, FBA = 7	(4) AB = 7, FBA = 7
	(5) D = 6, DF = 6

**Table 4 sensors-22-08063-t004:** Represents 1-itemsets, Top-K 1-itemsets and 2-itemsets.

Sno	1-Itemsets = Support
1	A = 8
2	B = 9
3	C = 4
4	D = 6
5	E = 3
6	F = 10
**Sno**	**Top-K 1-Itemsets = Support**
1	F = 10
2	B = 9
3	A = 8
4	D = 6
5	C = 4
**Sno**	**2-Itemsets = Support**
1	FB = 9
2	FA = 8
3	FD = 6
4	FC = 4
5	BA = 7
6	BD = 5
7	BC = 4
8	AD = 4
9	DC = 3
10	AC = 2

**Table 5 sensors-22-08063-t005:** Candidate and TK_IFIs_ List.

Sno	Candidate List	
1	F = 10	
2	B = 9	
3	A = 8	
4	D = 6	
5	C = 4	
**Sno**	**Top-K IFIs**	**Support**
1	F	10
2	B	9
3	A	8
4	D	6
5	C	4

**Table 6 sensors-22-08063-t006:** Candidate and TK_IFIs_ List.

Sno	Candidate List	
1	FB = 10	
2	FA = 9	
3	FD = 8	
4	FC = 6	
**Sno**	**Top-K IFIs**	**Support**
1	F	10
2	FB, B	9
3	FA, A	8
4	FD, D	6
5	FC, C	4

**Table 7 sensors-22-08063-t007:** Candidate and TK_IFIs_ List.

Sno	Candidate List	Pruning Strategies
1	FBA = 7	
2	FBD = 5 BD = 5	TIBCP
3	FBC = 4 C = 4	IBCP
**Sno**	**Top-K IFIs**	**Support**
1	F	10
2	FB, B	9
3	FA, A	8
4	FBA	7
5	FD, D	6

Backtrack on the node with Head Label F to extend with item A from the candidate list {A, D, C} [7]{ D, C}.

**Table 8 sensors-22-08063-t008:** Candidate and TK_IFIs_ List.

Sno	Candidate List	Pruning Strategies
1	FAD = 5 since AD = 4	TIBCP
2	FAC = 4 since C = 4	IBCP
**Sno**	**Top-K IFIs**	**Support**
1	F	10
2	FB, B	9
3	FA, A	8
4	FBA	7
5	FD, D	6

Backtrack on the node with Head Label F to extend with item D from the candidate list {D, C}, {FD} [13].

**Table 9 sensors-22-08063-t009:** Candidate and TK_IFIs_ List.

Sno	Candidate List	Pruning Strategies
1	FDC = 3 C = 4	IBCP
**Sno**	**Top-K IFIs**	**Support**
1	F	10
2	FB, B	9
3	FA, A	8
4	FBA	7
5	FD, D	6

**Table 10 sensors-22-08063-t010:** Datasets used for comparative analysis.

Sno	Database	Items	Avg Length	Database Type	Transactions
1	Chess	75	37	Dense	3196
2	Connect	129	43	Dense	67,557
3	Retail	16469	10.3	Sparse	88,162
4	T40I10D100K	1000	40	Dense	100,000
5	T10I4D100K	1000	10	Sparse	100,000

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
