# Peer review of "Efficient Top-K Identical Frequent Itemsets Mining without Support Threshold Parameter from Transactional Datasets Produced by IoT-Based Smart Shopping Carts"

_sensors, 2022, doi:10.3390/s22208063_

Round 1

Reviewer 1 Report

The paper presents Efficient Top-K Identical Frequent Itemsets Mining without 2 Support Threshold Parameter from Transactional Datasets 3 Produced by IOT based Smart Shopping Carts. Paper is well written and looks good. However, the followings are a few observations: 1. Abstract needs to be more precise and should have numerical outcomes mentioned. 2. Explain the working properly using Fig 1. 3. After each function, please add punctuation. 4. For Fig. 1, the font is strange, and the authors should adjust it. 5. In running text, mathematical symbols and values should be italic. 6. There are various works on IoT. For example A distributed ensemble design based intrusion detection system using fog computing to protect the internet of things networks, Design of anomaly-based intrusion detection system using fog computing for IoT network, P2IDF: a privacy-preserving based intrusion detection framework for software defined Internet of Things-fog (SDIoT-Fog) Discuss in related Sections. 7. There are some text and grammatical errors in the manuscript. It is recommended to read the full text and make corrections.

Author Response

Dear Reviewer(s) 

Thank you very much for the comments.

Regards 

Reviewer 2 Report

Research summary: 

The topic is relevant and may be of interest to a broad range of the journal's readers. However, this reviewer has some major concerns about the paper.

Major Strengths: The major strengths of the research are:

- The topic is interesting

- The proposed approach has been properly designed and developed

- The evaluation is interesting

Major Weaknesses: The major weaknesses of the research are:

- The structure and contents of the paper need to be improved.

- The differences between the proposed approach and the SOTA should be further discussed

Grammar and Readability:

The paper is well-written and clear. However, there are some typos that require to be reviewed.

Specific Comments: My specific comments concerning this manuscript are:

- The abstract does not highlight the specifics of the research or findings but contains too much background information. Some details of the research would be nice for example numbers addressing the sample, data, percentage improvement, etc.. Remove some of the background material and add some details of the research. Moreover, it is good to provide some specifics (e.g., sample size, dataset size, numbers from results, etc.).   

- There needs to be an explicit research objective(s) and/or research question(s) stated, preferably as a separate section. This helps readers find out what the research is trying to address.

- The introduction section is dispersive. It should focus on the challenges of the research area and the limitations of existing algorithms. However, the section often contains unnecessary digressions.

- "Many of algorithms have been proposed in the area of mining the topmost frequent itemsets" It is also necessary to introduce data profiling algorithms based on similar search strategies that are based on Apriori or similar strategies, such as https://doi.org/10.1109/TKDE.2020.2967722, https://doi.org/10.1145/2882903.2915203, https://doi.org/10.1109/ICDE.2019.00137

- There is a plethora of works that have addressed similar problems, but it is necessary to further highlight the novelty between the proposed study and the related literature.

- Are the datasets publicly available? If do, it is necessary to add URLs to access them in order to permit the reproducibility of the study. 

- Related Work is not very detailed and contains only a few related works. It is necessary to introduce a discussion on the works for the extraction of metadata from dynamic scenarios that exploit similar methodologies, such as https://openproceedings.org/2019/conf/edbt/EDBT19_paper_32.pdf, https://doi.org/10.1145/3397462, https://doi.org/10.14778/3401960.3401965

- The reference list needs tidying up, as there are references missing items or formatting issues. Please be consistent with the formatting and use some standard formatting style. 

- Figures 1, 2, 4 should be replaced with real pseudocodes.

- I suggest inserting a table summarizing the symbols used for discussing the proposed methodology.

- The size and quality of Figures 3 and 5 should be improved.

- The datasets involved in the evaluation seems too small given the sizes real-world applications currently need to deal with. Note that other solutions, also  cited and discussed in the related work, use larger datasets of 1.1 million rows.

- The comparative evaluation should consider multiple algorithms but the authors limit to only comparing a few datasets. Comparative evaluation with other algorithms is required.

Concluding Remarks:

The goal of the paper is interesting, but the authors should better discuss the novelty of the work. Moreover, it is necessary to perform a more detailed experimental evaluation, also considering larger real-world datasets.  

Round 2

Reviewer 1 Report

accept